# Serological Conversion through a Second Exposure to Inactivated Foot-and-Mouth Disease Virus Expressing the JC Epitope on the Viral Surface

**DOI:** 10.3390/vaccines11091487

**Published:** 2023-09-14

**Authors:** Seong Yun Hwang, Sung Ho Shin, Sung-Han Park, Min Ja Lee, Su-Mi Kim, Jong-Soo Lee, Jong-Hyeon Park

**Affiliations:** 1Animal and Plant Quarantine Agency, 177 Hyeoksin 8-ro, Gimcheon 39660, Republic of Korea; hsy8592@korea.kr (S.Y.H.); ikarus121@korea.kr (S.H.S.); shpark1124@korea.kr (S.-H.P.); herb12@korea.kr (M.J.L.); beliefsk@korea.kr (S.-M.K.); 2College of Veterinary Medicine, Chungnam National University, Daejeon 34314, Republic of Korea; jongsool@cnu.ac.kr

**Keywords:** foot-and-mouth disease, linear B-cell epitope, serological conversion, vaccine coverage, IL-12, IFNγ

## Abstract

Foot-and-mouth disease (FMD) is a fatal contagious viral disease that affects cloven-hoofed animals and causes severe economic damage at the national level. There are seven serotypes of the causative foot-and-mouth disease virus (FMDV), and type O is responsible for serious outbreaks and shows a high incidence. Recently, the Cathay, Southeast Asia (SEA), and ME-SA (Middle East-South Asia) topotypes of type O have been found to frequently occur in Asia. Thus, it is necessary to develop candidate vaccines that afford protection against these three different topotypes. In this study, an experimental FMD vaccine was produced using a recombinant virus (TWN-JC) with the JC epitope (VP1 140–160 sequence of the O/SKR/Jincheon/2014) between amino acid 152 and 153 of VP1 in TWN-R. Immunization with this novel vaccine candidate was found to effectively protect mice against challenge with the three different topotype viruses. Neutralizing antibody titers were considerably higher after a second vaccination. The serological differences between the topotype strains were identified in guinea pigs and swine. In conclusion, a significant serological difference was observed at 56 days post-vaccination between animals that received the TWN-JC vaccine candidate and those that received the positive control virus (TWN-R). The TWN-JC vaccine candidate induced IFNγ and IL-12B.

## 1. Introduction

Foot-and-mouth disease (FMD) is an important animal disease that is highly contagious in cloven-hoofed animals, such as cattle, pigs, and goats, and can threaten a nation’s livestock industry [1,2]. The foot-and-mouth disease virus (FMDV) is an RNA virus from the *Aphthovirus* genus belonging to the *Picornaviridae* family [3]. Since the FMDV is a single-stranded positive-sense RNA virus, it is highly mutated. The FMDV has seven serotypes and many topotypes, some of which can result in particularly serious economic losses [4]. Thus, FMD is one of the infectious animal diseases under the management of the World Organization for Animal Health (WOAH) [5].

Outbreaks of FMD in Taiwan in 1997 (due to a swine-derived virus), the United Kingdom in 2001, and the Republic of Korea in 2014 caused serious economic damage [6,7,8]. To minimize and prevent the damage caused by FMD, vaccination has been implemented in many countries where FMDV is currently found, including the Republic of Korea [9]. In addition, vaccination plays an important role in prevention strategies for FMD-free countries at a high risk of epidemic outbreaks [10].

There are five commonly known neutralizing antigenic sites on the FMDV: site 1 consists of amino acids (aa) 140–160 and 208 of VP1 (GH loop), site 2 consists of aa 70–77 (B–C loop) and aa 131–134 (E–F loop) of VP2, site 3 consists of aa 42–51 (B–C loop) of VP1, site 4 consists of aa 56–58 of VP3, and site 5 consists of aa 149 of VP1 [11,12]. To develop an innovative vaccine, we focused on site 1 (GH loop of VP1), a known linear B-cell epitope [13]. We selected site 1 because its immunogenic epitope can be modified without disrupting the icosahedral shape of the FMDV; it is even possible to insert an amino acid sequence that protrudes from the surface of the viral capsid in a linear manner [14]. Functionally, the antigenic site 1 induces a protective immunity [15,16,17].

In terms of detecting the host response to novel candidate vaccines, mouse FMD challenge models are used to rapidly and accurately confirm the ability of novel viruses and antigens to activate host defense mechanisms [18,19]. In addition, guinea pigs are used in serological analyses as an immunological experimental model for FMD [20,21].

We previously found in studies with recombinant vaccine strains that the P1 region was replaced by O/Taiwan/97 strain in the O1maisa backbone [22]. Although the TWN-R strain was effective as a vaccine strain, we tried to develop a vaccine strain matching more diverse topotypes than TWN-R by inserting the VP1 140–160 epitope of the O/SKR/Jincheon/2014 (JC) strain between amino acids 152 and 153 of VP1 in TWN-R. Here, the effect of adding the VP1 140–160 epitope of the JC strain was examined by comparing the immunogenicity of the TWN-JC recombinant virus with that of the positive control virus (TWN-R), which contained only P1 of the O/Taiwan/97 (TWN) strain.

## 2. Materials and Methods

### 2.1. Plasmid Preparation for Infectious Clones

The recombinant FMDV was prepared by inserting the VP1 140–160 amino acid sequence (the linear B-cell epitope) of the JC (O/SKR/JC/2014, GenBank No. KX162590.1) strain of SEA topotype into the previously studied TWN-R strain of the Cathay topotype [22]. The following oligonucleotide primers were used to insert the epitope into the TWN P1 gene: JC_epitope_F 5′-GCTGGCACCGAAAGCGGCGAGGCCATTGCCCCAGAAGGCAGAAAGAACTC-3′ and JC_epitope_R 5′-ACTTGGAGATCGCCTCTCACGTTGGGCAGTGAAGCTAACACTTGAAGGTC-3′. The 3B region was mutated as a differentiating infected from vaccinated animals (DIVA) marker in the recombinant virus [23]. Mutations in the 3B region replaced 3B_1_B_2_ with a double repeat of 3B_3_ in the viral genome to develop a vaccine strain.

### 2.2. Cell Culture and Recovery of Viral Particles

BHK-T7-9 cells [24] (cells expressing T7 RNA polymerase) were maintained in Glasgow Minimum Essential Medium (Gibco BRL, Paisley, UK) supplemented with 5% fetal bovine serum (Gibco BRL, Paisley, UK) and 10% tryptose phosphate broth (Sigma-Aldrich, St. Louis, MO, USA). ZZ-R 127 cells (Fetal goat tongue cell line) were maintained in Dulbecco’s Modified Eagle’s Medium/F12 (Corning, Union City, NJ, USA).

Recombinant viruses were recovered using the procedures described in a previous study [25], and the prepared plasmid was transfected with the linearized plasmids using Lipofectamine 3000 (Invitrogen, Carlsbad, CA, USA) to generate the chimeric viruses in the BHK-T7-9 cells. After being transfected and incubated for 48–72 h at 37 °C, the recombinant virus was harvested via three freeze–thaw cycles, and the harvested virus was used to infect fresh ZZ-R 127 cells. All the experiments involving the FMDV or virus-infected animals were conducted in a biosafety level 3 (BSL-3) facility at the Animal and Plant Quarantine Agency (APQA).

### 2.3. Measurement of Viral Growth

Each type of virus was cultivated in each of the three cell types (ZZ-R 127; goat tongue cell line, LFBK; porcine kidney cell line, and BHK-21; baby hamster kidney cell line), and a viral growth curve was generated for each case. Cell monolayers in 12-well plates were infected with the TWN-R or TWN-JC virus at a multiplicity of infection (MOI) of 0.01 or 0.001 and incubated at 37 °C in 5% CO_2_. After 1 h, the supernatant was replaced with fresh medium. RNA purification was performed by harvesting the supernatant at 0, 2, 4, 8, 12, 18, and 24 h post infection (hpi). A One-step Prime-script RT-PCR kit (Bioneer Inc., Daejeon, Republic of Korea) and a CFX96 Touch Real-time PCR Detection system (Bio-Rad, Hercules, CA, USA) were used to generate the RNA-based data that were utilized to produce the viral growth curves. The experiments were repeated at least three times.

### 2.4. Purification for Viral Capsid Protein

The inactivated FMDV antigens were purified as described in a previous study, with some modifications. Briefly, suspended BHK-21 cells were maintained in CD BHK-21 Production Medium (Gibco BRL, Paisley, UK) in a 5% CO_2_ atmosphere. The virus grown in the suspended BHK-21 cells was harvested by freezing and thawing when a complete cytopathic effect was observed. The cell debris was removed after the inactivation of the virus using binary ethyleneimine (0.003 N) at 26 °C for 24 h. The inactivated virus in the clarified culture was subsequently precipitated by incubating it with 7.5% PEG 6000 and 2.3% NaCl overnight at 4 °C, and the pellet was resuspended in Tris-KCl buffer to a final concentration of 200 times. The resuspended pellet was subsequently purified via centrifugation through a 15–45% sucrose gradient in Tris-KCl buffer at 30,000 rpm for 4 h at 4 °C using an SW41 rotor. The content of the pellet was determined via a spectrophotometric analysis at 259 nm. The inactivated antigens of the recombinant virus were identified using the FMDV Ag-lateral flow assay kit (Princeton Bio Meditech Corporation, Monmouth Junction, NJ, USA). The purified antigens were separately adsorbed onto carbon-coated copper grids and visualized under a transmission electron microscope (Hitachi H7100FA, Tokyo, Japan).

### 2.5. Vaccination and FMDV Challenge in C57BL/6 Mice

The candidate vaccine consisted of the antigen and adjuvant (ISA 206; 50%, Al(OH)_3_; 10%, saponin; 1%). Mice were vaccinated via an intramuscular injection, and the mice in the experimental group received 1/10 (1.5 μg), 1/40 (0.375 μg), 1/160 (0.0937 μg), or 1/640 (0.0234 μg) of 15 μg (pigs were vaccinated with a 15 μg/dose). Seven days after vaccination, the mice were challenged with a virus (administered via an intraperitoneal injection), and their survival rates and body weights were observed for a further seven days. Seven-week-old C57BL/6 female mice (*n* = 5; Orient Co. Ltd., Seoul, Republic of Korea) were vaccinated and then challenged with the ME-SA topotype (O/Vietnam/2013; VIT) at a dose of 3 × 10^4^ TCID_50_/0.1 mL. To induce a lethal condition, IFNαR K/O mice (*n* = 3 or 4; Watson RnD, Gyeonggi-do, Republic of Korea) were vaccinated and then challenged with the SEA (O/SKR/Jincheon/2014; JC) or Cathay (O/Taiwan/97; TWN) topotype at a dose of 1 × 10^5^ TCID_50_/0.1 mL. All the animals were kept at the Animal and Plant Quarantine Agency (APQA) and were used with the approval of the Animal Care and Use Committee.

### 2.6. Determination of Immune Responses via ELISA, VNT, and Calculation of Serological Relationships in Guinea Pigs and Swine

Female guinea pigs (*n* = 5; 250 g) were vaccinated intramuscularly with a 7.5 μg/dose of antigen mixed with an ISA 206, saponin, and aluminum hydroxide gel adjuvant. They received a booster dose at 21 dpv. Blood was drawn at 0, 14, 28, and 42 dpv. Eight- to ten-week-old pigs (*n* = 5) were vaccinated with a 15 μg/dose of antigen mixed with an ISA 206, saponin, and aluminum hydroxide gel adjuvant. Pig serum was collected at 7, 14, 21, 28, 42, and 56 dpv. The pigs received a booster dose at 28 dpv. Serum antibodies specific to the FMDV structural proteins (SP) were detected via ELISA using PrioCheck FMDV O (Prionics AG, Schlieren-Zurich, Switzerland). When a sample returned a percent inhibition value of > 50%, the respective animal was regarded as having demonstrated an immune response.

The serum samples were heat-inactivated at 56 °C for 30 min. The cell density was adjusted to form a 70% monolayer, and two-fold serial dilutions of the serum samples were prepared. The diluted serum samples were then incubated with 100 TCID_50_ of the FMDV (including O/SKR/Boeun/2017; BE) for 1 h at 37 °C. After 1 h, an LFBK (porcine kidney) cell suspension was added to the wells of 96-well plates. After two to three days, the extent of the cytopathic effect (CPE) in each well was recorded to determine the neutralization titer, which was calculated as log10 of the reciprocal antibody dilution required to neutralize 100 TCID_50_ of the virus.

### 2.7. 3D Structural Model Analysis

The crystal structure of the O PanAsia strain (PDB accession no. 5NE4) was used as a template for modeling the capsid protein (protomeric subunit) of the candidate vaccine strains, TWN-R and TWN-JC, using SWISS-MODEL. The localization of the inserted epitope region in the protomeric subunits and an epitope analysis were performed using the Pymol molecular graphics system (Version 2.3.4, Schrodinger LLC, New York, NY, USA).

### 2.8. Cytokine ELISA

The ELISAs used to detect the porcine IFNγ, IL-2, IL-12B (Cloud-Clone Corporation, Houston, TX, USA), IL-7, and IL-15 (Wuhan Fine Biotech Co., Wuhan, China) in the serum were performed according to the manufacturer’s instructions.

### 2.9. Statistical Analyses

Data are presented as means ± standard deviations (SD) and represent the results from at least three independent experiments. Differences between the groups were determined by performing an analysis of variance (ANOVA), and the means were compared using the Student’s *t* test. *p* values of * *p* < 0.05 and ** *p* < 0.01 were regarded as significant or highly significant.

### 2.10. Ethics Statement

The animal experiments were performed in strict accordance with the Animal and Plant Quarantine Agency (APQA)’s guidelines for the care and use of laboratory animals. All the procedures that involved animals were approved by the Institutional Animal Care and Use Committee of the APQA of South Korea (approval no. 2022-694). All efforts were made to minimize animal suffering.

## 3. Results

### 3.1. Identification of Candidate Vaccine Strains and Viral Capsid Purification

TWN-R contains the TWN strain’s P1 sequence in the O1manisa backbone, and TWN-JC was constructed by inserting amino acid 140–160 from VP1 of O/SKR/JC/2014 (JC) between amino acid 152 and 153 of VP1 in TWN-R (Figure 1). It was predicted that the three-dimensional structure changed due to the addition of the JC epitope (linear B-cell epitope of O/SKR/JC/2014; VP1 140-160 amino acid). As a result of comparing the prediction models of TWN-JC and TWN-R, it was found that the outwardly protruding linear epitope of VP1 was changed (Appendix A). Transmission electron microscope (TEM) images showed no difference in the viral particle shapes and sizes between TWN-R and TWN-JC. It was confirmed that the inserted JC epitope sequence was maintained in the fourth suspension’s cell passage (Appendix A). To confirm whether there was a difference in the virus growth, the TWN-R and TWN-JC viruses were, respectively, infected with ZZ-R 127, LFBK, and BHK21 cells, and the virus replication was measured over time. There was no significant difference observed in the viral replication between the two virus strains (Figure 1 and Appendix A). Comparing the morphology of the virus in the TEM images and the viral replication, TWN-R and TWN-JC were similar. Therefore, it is considered that there was no problem with the virus stability due to the addition of the JC epitope.

### 3.2. Determination of the Protective Value of the Experimental Vaccine in Mice

To determine whether the addition of the JC strain’s epitope enhanced the immunogenicity of TWN-JC in comparison to TWN-R (positive control), the vaccinated mice were challenged with each virus topotype (Figure 2). When the mice were infected with the foot-and-mouth disease virus, there were mild symptoms such as slowed movement, loss of appetite, weight loss, and, in severe cases, death. In the O/SKR/JC/2014 and O/TWN/97 challenge group, IFNαR knock-out mice were used to clearly observe the differences in immunogenicity between the TWN-R and TWN-JC group. A survival rate of 0% was confirmed in the non-vaccinated group (negative control), while the survival rate was confirmed in proportion to the antigen concentration in the vaccinated group. A challenge with O/SKR/JC/2014 (SEA; Mya-98 lineage) or O/VIT/2013 (ME-SA; Pan-Asia lineage) showed that the vaccination with TWN-JC produced a significantly higher protection than that with TWN-R, indicating that TWN-JC has a high protective capability.

The mice vaccinated with TWN-JC and challenged with O/TWN/97 (Cathay topotype) were found to have the same survival rate at seven days post-challenge (dpc) as the mice vaccinated with TWN-R and challenged with the Cathay topotype. However, the mice vaccinated with a 1/40 dose (porcine vaccine dose) of TWN-R showed weight loss, whereas those vaccinated with TWN-JC showed no weight loss (Appendix A).

### 3.3. Immunogenicity of the Candidate Vaccine in Guinea Pigs

To determine the effect of the included epitope from the JC strain on the immunogenicity of the candidate vaccine, the serological difference between the animals vaccinated with TWN-R and TWN-JC was determined in the guinea pigs (Figure 3). After the first vaccination with either TWN-R or TWN-JC, blood was collected every two weeks and boosting was performed at 21 dpv. The ELISA and VNT results showed that the vaccination with TWN-R induced slightly higher antibody levels at 14 dpv than the vaccination with TWN-JC; however, this situation was reversed after boosting. Serum samples from the animals vaccinated with TWN-R were found to have higher levels of neutralizing antibodies against the Cathay topotype than those from the animals vaccinated with TWN-JC, due to the high homology in the viral structures. However, the serum samples from the animals vaccinated with TWN-JC were found to have higher levels of neutralizing antibodies against O/SKR/JC/2014 (SEA; Mya-98 lineage), O/VIT/2013 (ME-SA; Pan-Asia lineage), and O/SKR/BE/2017 (ME-SA; Ind2001 lineage) than those from the animals vaccinated with TWN-R.

### 3.4. Evaluation of Neutralizing Antibodies Generated in Vaccinated Pigs

After confirming the effect of the JC epitope on the immunogenicity of the candidate vaccine in the guinea pigs, the immunogenic effect of the JC epitope was determined in five pigs, where pig was the target animal. Boosting was performed 28 days after the first vaccination, and ELISAs and VNTs were used to measure the resultant antiviral antibody levels (Figure 4). The results were similar to those found with the vaccinated guinea pigs.

Priming and boosting the pigs with TWN-R induced more neutralizing antibodies effective against the Cathay topotype than priming and boosting them with TWN-JC. However, higher anti-SEA and anti-ME-SA neutralizing antibody levels were generated by priming and boosting with TWN-JC in comparison to that with TWN-R. A serological analysis performed using guinea pigs and porcine serum samples showed that TWN-R demonstrated a higher correlation than TWN-JC, and that TWN-JC demonstrated a higher correlation with O/SKR/JC/2014, O/VIT/2013, and O/SKR/BE/2017.

To confirm the broad immunogenicity of TWN-JC compared to TWN-R, we analyzed VN titers before and after boosting. Before boosting, it was found that TWN-R and TWN-JC had a similar serum neutralizing antibody level. After boosting with TWN-JC, the neutralizing antibodies against the SEA and ME-SA topotype viruses significantly increased compared to those with TWN-R (Figure 5).

### 3.5. Effects of the JC Epitope on Cytokine Secretion

Cytokines were also detected via ELISA to confirm whether the addition of the JC epitope affected the cytokine secretion (Figure 6). Cytokine secretion was observed using the 0 dpv (pre-vaccination), 7 dpv, 28 dpv (before boosting), and 56 dpv (after boosting) serum of the immunized pigs. All the cytokine levels in both TWN-R and TWN-JC increased initially after vaccination and then tended to decrease over time by being normalized with pre-vaccination serum. The animals vaccinated with TWN-JC demonstrated higher IFNγ and IL-12B levels than those vaccinated with TWN-R at 7 dpv. The levels of IFNγ and IL-12B showed an approximate two-fold difference, but these results were not statistically significant. The IL-2, IL-7, and IL-15 levels also increased at 7 dpv, but there was no significant difference between TWN-R and TWN-JC.

## 4. Discussion

The recent SARS-CoV-2 pandemic has increased the interest in single-stranded RNA viruses and reinforced the fact that effective vaccines are required to prevent infectious viral diseases [26]. Most FMDV strains currently used in vaccines have limitations; for example, they tend to elicit short-lived antibodies in pigs or only induce humoral immunity [27]. To develop vaccine strains with excellent protective effects, two main paths are followed: the generation of attenuated viruses with a reduced pathogenicity and the production of epitope-based vaccines [28,29,30,31,32].

A considerable amount of research has been conducted to develop B- and T-cell epitope-based vaccines that protect against diseases mediated by RNA viruses [33,34,35], including FMD [36]. Here, we selected a universal B-cell epitope from the VP1 of the FMDV and designed a multi-epitope virus strain with the aim of increasing the number of neutralizing antibodies elicited that were effective against three different topotypes [37].

The results of this study suggest that the novel recombinant strain (TWN-JC) can be used as a vaccine strain because the surface structure of the virus particle is changed, but the replication of TWN-JC was not significantly different compared to TWN-R (Figure 1 and Appendix A). Furthermore, the immunogenicity of the recombinant virus in mice was improved by the addition of the linear B-cell epitope from the JC strain; the mice that received the TWN-JC candidate vaccine generated more anti-viral antibodies that were effective against the SEA and ME-SA topotypes than the mice that received TWN-R. When assessed against the Cathay topotype, TWN-R and TWN-JC were found to produce the same survival rate; however, the mice that received TWN-JC showed less weight change than those that received TWN-R (Figure 2 and Appendix A). In both the TWN-R and TWN-JC groups, the survival rate was higher in the O/VIT/2013 (ME-SA; Pan-Asia lineage) challenge group than in the O/TWN/97 or O/SKR/JC/2014 group. The reason for this is that the Pan-Asia lineage strain among ME-SA is thought to be antigenically highly reactive to most type O viruses [38].

Compared to TWN-R, TWN-JC was also found to induce greater protection in the mice against the SEA and ME-SA topotypes and a potent immune response in the guinea pigs that was effective against the SEA and ME-SA topotypes. It was noted that the immune response induced by TWN-JC was relatively less effective against the Cathay topotype than that induced by TWN-R. This was because the viral capsid structure in TWN-JC was changed due to the addition of the JC epitope (Figure 3 and Appendix A).

A serological analysis performed using pig serum returned similar results to those observed in the experiments with the guinea pigs. Prior to receiving a booster dose, the animals vaccinated with TWN-JC showed relatively low neutralizing antibody levels compared to those vaccinated with TWN-R. However, after boosting, they were found to have relatively high titers. The vaccine matching was analyzed using porcine serum and each topotype. At 28 dpv, TWN-R and TWN-JC observed similar neutralizing antibody levels for the three topotypes, but at 56 dpv, TWN-R showed a high VN titer only with the Cathay topotype, whereas TWN-JC confirmed a high VN titer with the three topotypes (Figure 4 and Figure 5).

Our findings indicate that adding the linear B-cell epitope from the JC strain to the GH loop region of VP1 resulted in the induction of high neutralizing antibody levels up to 56 dpv that could neutralize a range of viruses [31]. The addition of the epitope appeared to have a positive effect on the production of neutralizing antibodies, inducing a gradual rise in their quantity and an enhanced immune response against the tested viruses. A similar phenomenon (i.e., the induction of long-lasting antibodies) was reported in a previous comparative study conducted with chimeric viruses [27,38].

To identify the reason that TWN-JC induced such a robust immune response after boosting, cytokine secretion was measured (Figure 6). Given that the IFNγ and IL-12B secretion was increased in the animals that received TWN-JC compared to the animals that received TWN-R, it is reasonable to conclude that the presence of the JC epitope stimulated the IL-12 and IFNγ levels [39]. IFNγ modulates the immune response to viral infection and affects memory T cells [27,40,41,42,43,44]. Furthermore, IFNγ is known to be involved in the host defense response in pigs, and cellular immunity is induced by IFNγ and IL-12, which mediate protection against the FMDV [45,46].

## 5. Conclusions

Despite the insertion of the JC epitope, the virus was stably formed, and it was confirmed that immunogenicity against the three topotypes (Cathay, SEA, and ME-SA topotype) was equally improved in the mice, guinea pigs, and swine depending on the presence or absence of the JC epitope. As the JC epitope was additionally inserted into antigenic site 1, the vaccine matching rate for the three topotypes increased. It is considered that serological conversion occurred after the animals received a booster dose and that memory-related immunity was enhanced by the addition of the JC epitope.

## Figures and Tables

**Figure 1 vaccines-11-01487-f001:**
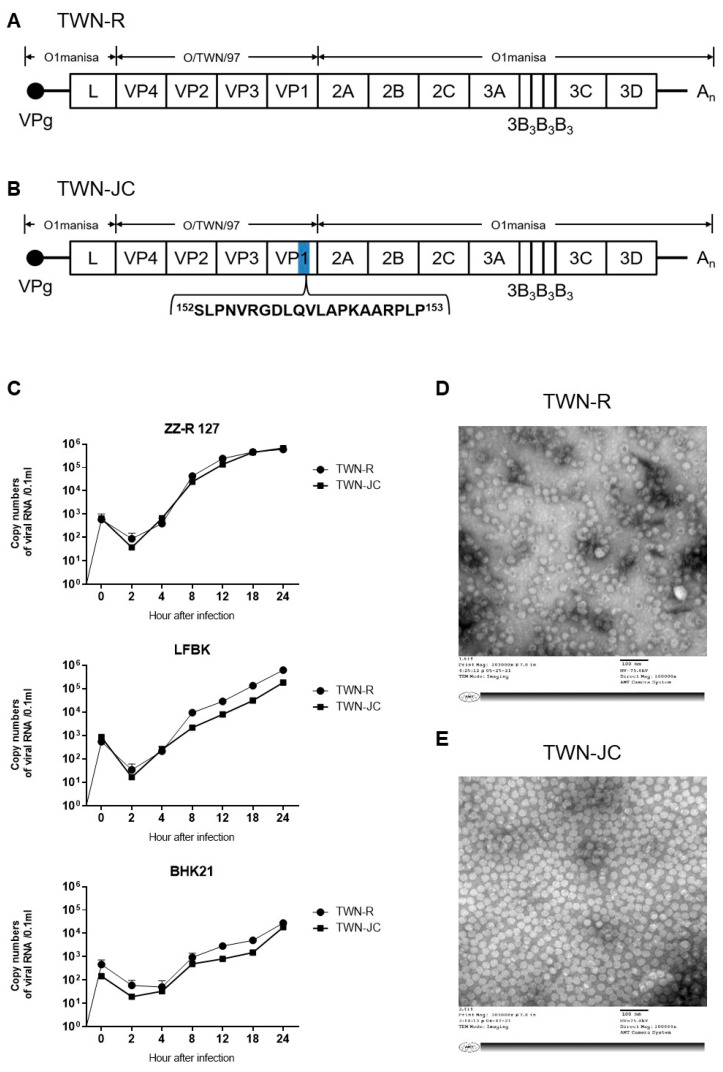
Characteristics of the TWN-R and TWN-JC. Schematic diagram of the TWN-R and TWN-JC genomes (**A**,**B**). TWN-JC is a novel vaccine candidate that has a JC epitope inserted into the TWN-R strain. TWN-R, which has the TWN P1 genome and O1manisa backbone, and TWN-JC, which has the VP1 140–160 linear B-cell epitope of the JC strain inserted into the VP1 GH loop region of TWN-R. Comparison of viral replication between TWN-R and TWN-JC in vitro. Three types of cells (ZZ-R 127, LFBK, and BHK-21) were infected at multiplicity of infection (MOI) of 0.01, and virus growth was measured at 0, 2, 4, 8, 12, 18, and 24 h post-infection (**C**). Transmission electron microscope (TEM) imaging of TWN-R (**D**) and TWN-JC (**E**).

**Figure 2 vaccines-11-01487-f002:**
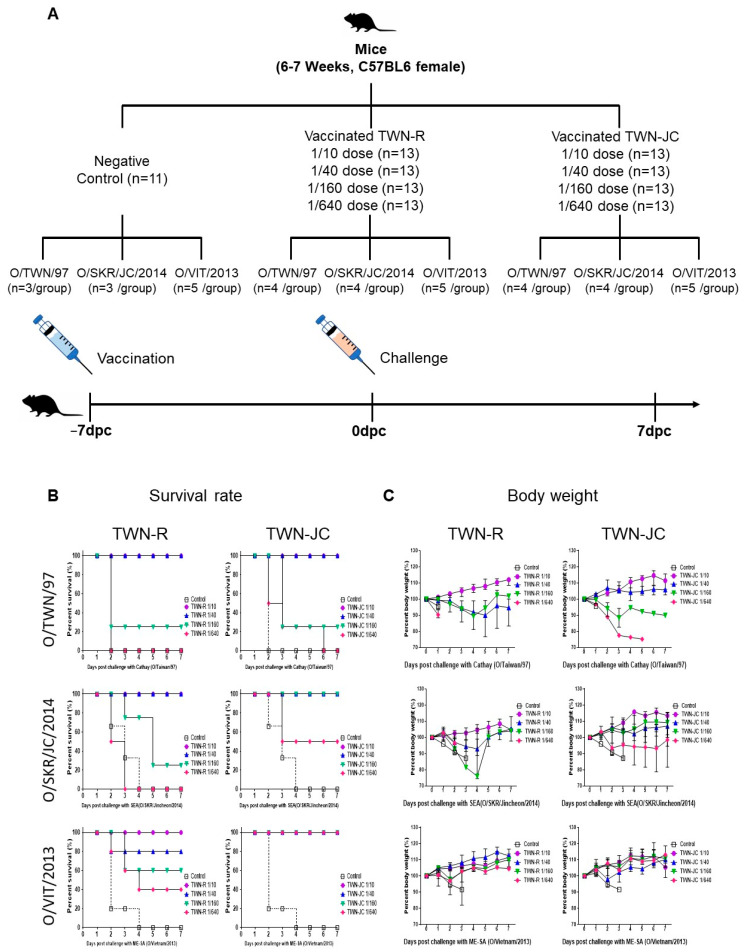
Protection against serotype O topotype viruses mediated by vaccination with TWN-R or TWN-JC in mice. Mice were vaccinated with TWN-R or TWN-JC at 1/10, 1/40, 1/160, or 1/640 of the dose used in pigs and challenged with three virus topotypes (ME-SA, SEA, and Cathay) at 7 dpv (**A**). The survival rate (**B**) and body weight (**C**) of the mice were measured for a further 7 days post-challenge.

**Figure 3 vaccines-11-01487-f003:**
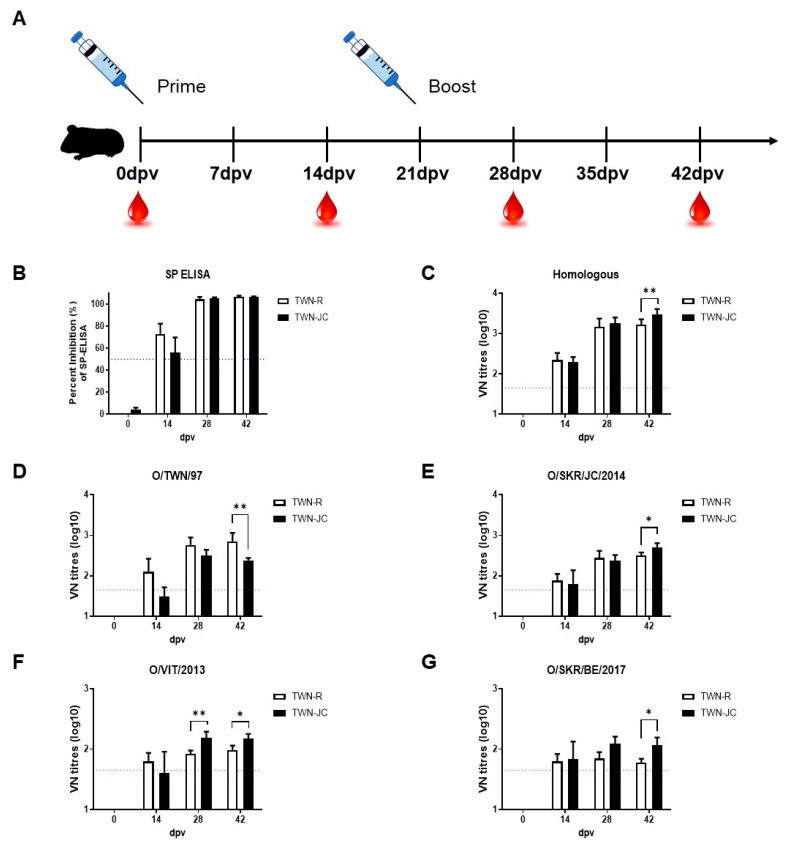
Immunogenicity of the recombinant virus in guinea pigs. Guinea pigs (*n* = 5) were vaccinated with 7.5 µg of inactivated antigen mixed with ISA 206, aluminum hydroxide gel adjuvant, and saponin. Sera were collected at 0, 14, 28, and 42 dpv, and the animals were boosted at 21 dpv. Schematic diagram of the experimental strategy (**A**), Structure Protein (SP) ELISA results generated using serum from guinea pigs vaccinated with TWN-R or TWN-JC (**B**), and the homologous (**C**) and heterologous neutralizing antibody titers against TWN (**D**), JC (**E**), VIT (**F**), and BE (**G**) in guinea pigs vaccinated with TWN-R or TWN-JC. A percent inhibition (PI) value > 50 was considered the cutoff for a positive reaction (blue dotted line). VN titers (log10) > 1.65 were regarded as positive (red dotted line). *p* values of *p* < 0.05 (*) and *p* < 0.01 (**).

**Figure 4 vaccines-11-01487-f004:**
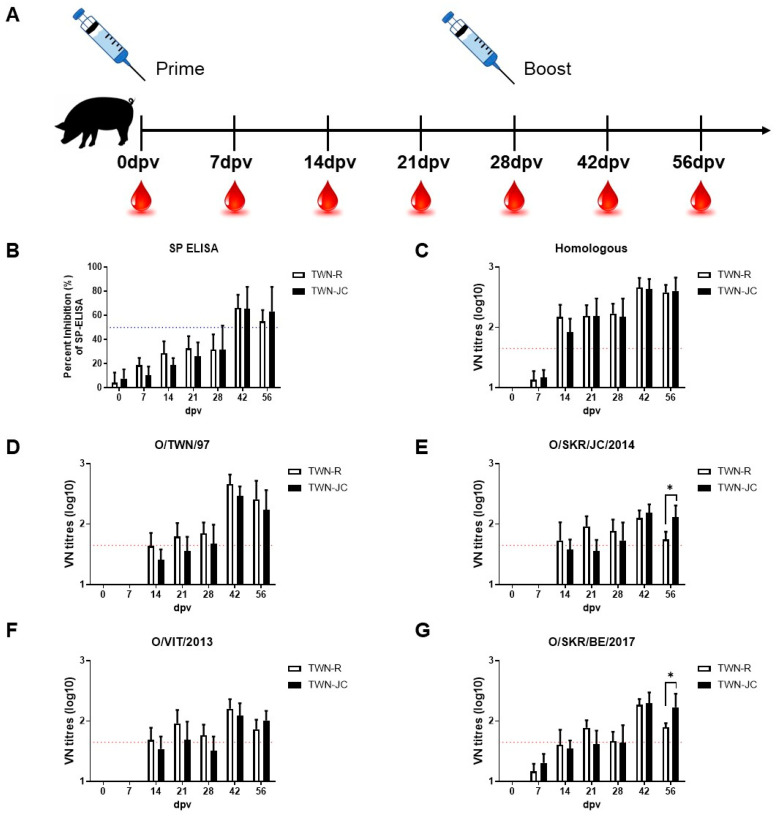
Immunogenicity of the recombinant virus in pigs. Pigs (*n* = 5) were vaccinated with 15 µg of inactivated antigen mixed with ISA 206, aluminum hydroxide gel adjuvant, and saponin. Sera were collected at 0, 7, 14, 21, 28, 42, and 56 dpv, and the animals were boosted at 28 dpv. Schematic diagram of the experimental strategy (**A**), Structure protein (SP) ELISA results generated using serum from pigs vaccinated with TWN-R or TWN-JC (**B**), and the homologous (**C**) and heterologous neutralizing antibody titers against TWN (**D**), JC (**E**), VIT (**F**), and BE (**G**) in pigs vaccinated with TWN-R or TWN-JC. A PI value > 50 was considered the cutoff for a positive reaction (blue dotted line). VN titers (log10) > 1.65 were regarded as positive (red dotted line). *p* values of *p* < 0.05 (*).

**Figure 5 vaccines-11-01487-f005:**
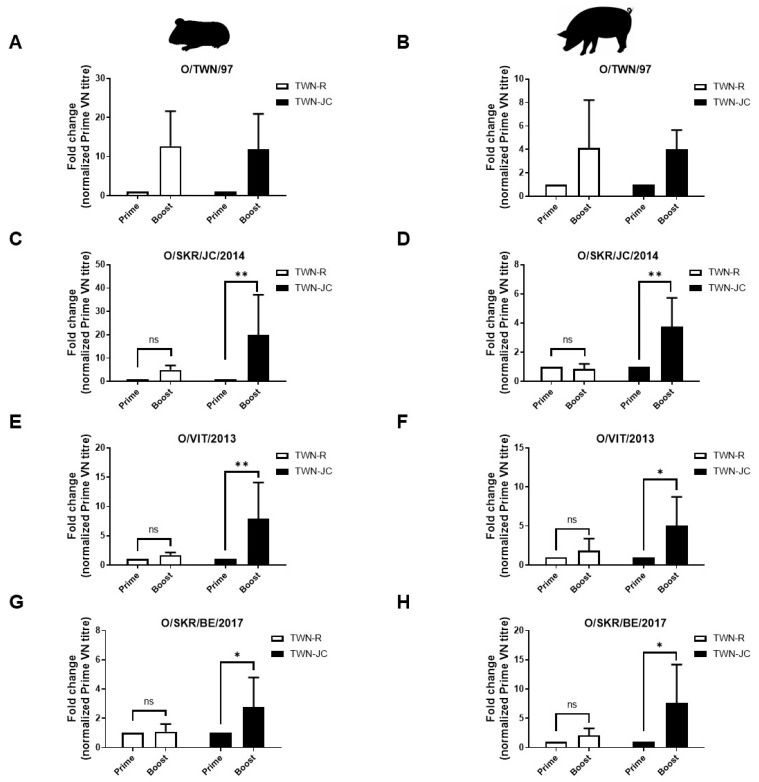
Comparison of specific serological changes associated with the different topotypes in guinea pigs (*n* = 5) and pigs (*n* = 5). The fold change associated with the different topotypes were compared using the prime (14 dpv) and boost (42 dpv) serum samples obtained from the guinea pigs (**A**,**C**,**E**,**G**) and the prime (28 dpv) and boost (56 dpv) serum samples obtained from the pigs (**B**,**D**,**F**,**H**). The fold change of the boost VN titer was normalized to prime VN titer. *p* values of *p* < 0.05 (*) and *p* < 0.01 (**).

**Figure 6 vaccines-11-01487-f006:**
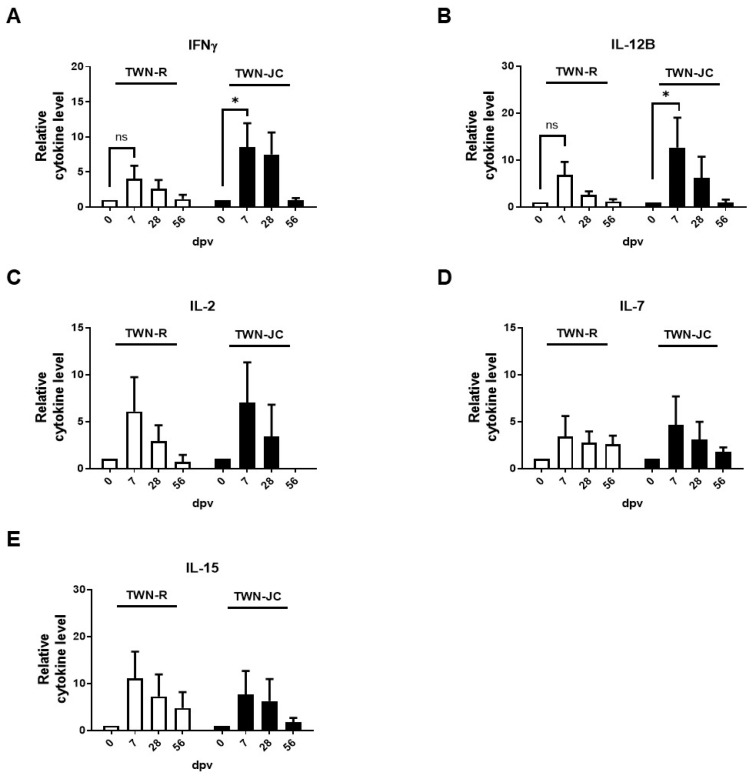
Cytokine secretion in pigs (*n* = 5) vaccinated with TWN-R or TWN-JC. The serum levels of IFNγ (**A**), IL-12B (**B**), IL-2 (**C**), IL-7 (**D**), and IL-15 (**E**) were detected post-vaccination using an ELISA. The 7, 28, and 56 dpv serum samples obtained from pigs immunized with TWN-R or TWN-JC were tested and normalized to serum before vaccination. *p* values of *p* < 0.05 (*).

## Data Availability

All datasets generated for this study are included in the article.

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
