# Peer review of "Serological Conversion through a Second Exposure to Inactivated Foot-and-Mouth Disease Virus Expressing the JC Epitope on the Viral Surface"

_vaccines, 2023, doi:10.3390/vaccines11091487_

Round 1

Reviewer 1 Report

The manuscript entitled “Serological conversion through a second exposure to inactivated foot-and-mouth disease virus expressing a neutralizing epitope on the viral surface” by Heang et al. reports the immunogenicity of the inactivated reverse genetic derived, SEA-B cell epitope inserting-FMDV vaccine in guinea pigs and pigs. A B cell epitope of FMDV topotype SEA (JC strain) was inserted in the G-H loop of FMDV topotype Cathay (TWN-R) P1 resulting in TWN-JC P, which was subcloned into FMDV O1 Manisa backbone to generate TWN-JC infectious clone. The rescued TWN-TJ and TWN-R were produced in BHK-21 suspension cells to prepare antigen for vaccine production. The viruses were inactivated, concentrated, and purified before subjecting to vaccine formulation. Vaccination with the TWN-JC vaccine followed by homologous and heterologous (Cathay, SEA and ME-SA) challenges in a mice model showed comparable results with the TWN-R vaccines in the homologous challenge, while induced more potent protection against the heterologous challenges. Guinea pigs and pigs were immunized with the TWN-R and TWN-JC vaccines using the prime-boost regimens (TWN-R/TWN-R and TWN-JC/TWN-JC) and the sera from the immunized animals were examined by ELISA and serum neutralization test (SNT). The conclusive results were in accordance with the mice model. This study contains some interesting results. However, the experimental designs might be problematic, or the methods described in this manuscript were not clear.

Here are comments:

1.       The major concerns are that the control groups which are crucial for the reliable results were missing from all animal experiments, especially the vaccination and challenge experiments.

1.1   Without appropriate controls, it is hard to count on the results. In fact, the non-vaccinated, non-challenge control would ensure that no other factors interfered with the experiment, while the non-vaccinated, challenge control would confirm that the challenge viruses could induce the FMDV specific clinical signs. The non-vaccinated, non-challenge group is present in Figure 2; however, no such information in the methods. The authors must provide the results of non-vaccinated, challenge control group in the mice experiment. Otherwise, the method, results and discussion related to this experiment should be omitted from the manuscript.

1.2   If the mouse experiment will be included in the manuscript, the method of topic 2.5 (Vaccination and FMDV challenge in mice model) should be reorganized and rewritten to include the following information.

-Put the reference(s) that is the original mouse model for FMDV and whether or not it was modified in this study. How it was modified. Actually, briefly describe regarding the previous published mouse model of FMD in the Introduction section would be helpful.

-Which mice breeds were used in the study? How old were they?

-How many mice in total were used in this experiment and number of mice in each group.

-How were many groups of the treatments and controls included in this experiment? What were they?

-What were the antigens used in the vaccines? Were the same as that explained in topic 2.4 (Virus purification and capsid)? How much antigen was in each dose?

-What were the viruses used for the challenge? Include the Isolate names, lineages, topotypes and serotypes. How much concentration of each virus was used for the challenge? Why the dose of O/VIT/2013 was different from those of the other viruses [SEA (JC) and Cathay (TWN)]? Why the IFNa R knock out mice were used for just the SEA (JC) and Cathay (TWN) viruses?

-Briefly describe the FMDV specific signs in mice? What were case definition of FMD in mice and the criteria for making decision that the mice were sick/die/protected from FMDV.

In the Results section, number of surviving mice/total mice in each group should be reported.

-For the protective dose (PD50), the authors should change to mouse-PD50 thorough the manuscript. The authors should explain how to calculate the mouse-PD50 and the interpretation.

-Table 1: Include one more column at the far left with the topic “Topotype” and change the consecutive right column from “Topotype” to “Challenge strain”. The numbers of mouse-PD50 did not match with the corresponding numbers in the main text. Please check carefully and correct them.

1.3   The control groups are also important for immunized studies. It would be more reliable if the results of control groups in the guinea pig and pig experiments are included, which the Figures could be presented as Supplements. However, the results of immunity induction could be compared with the pre-immune serum, which is usually collected on day 0 before the 1st immunization, which is shown to be at the background level or negative.

-How much antigen was used in each dose of vaccines for the guinea pig vaccination.

-In the Results section and Figure captions, the word “inoculated” should be replaced by “vaccinated” when mentioned to vaccination with vaccines.

-Figure 3: The labels of virus strains (Topotypes) used for challenging appearing on the top of pictures D-G are too close to the upper pictures. These labels are not clear whether they belong to the upper or lower pictures. This should be adjusted.

2.       Strong evidence of TWN-JC infection is required to convince that modification of the G-H loop did not inhibit viral attachment and entry. The easiest method to show that TWN-JC is infectious is immunofluorescent or immunoperoxidase monolayer assay. The authors should show that an antibody specific to the B cell epitope could bind to and produce intracytoplasmic signals in TWN-JC infected cells but not TWN-R infected cells.

3.       Topic 2.2 should be rewritten and reorganized as it was confusing. Normally, the plasmid containing an infectious clone was transfected to the BHK-21 cells. The viruses generated in the cells were rescued and used for inoculating in the ZZ-R 127 cells. However, the authors started with virus rescue.

 “the infectious recombinant viral particles present in the BHK-T7-8-9 cells were transfected with the linearized plasmid using Lipofectamine 3000” This sentence should be expanded to clarify the following questions.

-What are the infectious recombinant viral particles present in the BHK-T7-8-9 cells? How is it important to produce infectious viruses from the clone cDNA?

-What were the chimeric viruses? TWN-R and TWN-JC?

- Which B cell epitope was used in this study. Is it on VP1, VP2 or VP3? Where does it present in the viral protein (indicate as nucleotide or amino acid position of the epitope on the viral protein). The information is present in the caption of Supplementary Figure 1, which should be in the method under Topic 2.2 and Figure S1 should be cited in the main text.

-In Figure S1, change picture A, B and C to C, A and B, respectively, and reorder the content in the Figure caption accordingly.

4.       Topic 2.3: In the third line of the paragraph, “…the TWN-R or TWN-JC virus at a MOI of 0.01 or 0.001…” is unclear. Please revise and cite Figure S2. How many replicates were there in this experiment?

5.       Topic 2.9: Did the data was tested for normality before using Student’s T-test for the analysis.

6.       Topic 3.1: Explain the electron microscopy results in the main text regarding size and morphology of the particles of both viruses.

7.       Topic 3.4: In the second – third lines of the first paragraph, “…in a target animal: the pig” should be changed to “…5 pigs, which pig is the target animal.”

-The second paragraph should be rewritten to explain the results in Figure 4. The last sentence should be omitted as it has no meaning.

-The result of serological relationship in the third paragraph should be reported as r value of vaccine matching.

8.       Topic 3.5: The cytokine levels at day 0 prior to vaccination should be displayed in Figure 6.

9.       Discussion:

-In the third paragraph from the end, the word “long lasting” should be removed because the last days of the experiments in this study were 42 and 56 days.

-In the second paragraph from the end, “…IL12/IFNg axis [38]” should be changed to “…IL12/IFNg level [38]”

-The authors should discuss how the G-H loop of FMDV VP1 could harbor foreign amino acids, regarding the 3D structure, possibility and constraint of amino acid addition in the G-H loop in terms of number and type of amino acids. Also describe the 3D structures predicted in this study. Where is the tentative location of the modified G-H loop on the virus particle? How stability it is?

-The modified G-H loop of TWN-JC seemed to extruded out of the receptor binding site. How did it effect on the binding of the virus to its receptor?

10.   Miscellaneous corrections such as…

-Rewrite the sentence in lines 33-36.

-Line 40: Change “…being conducted…” to “…implemented…”

-Lines 42-43: Change “…of countries without FMD but with high risk of the outbreaks [10].” to “…of FMD free countries at high risk of the epidemic outbreaks [10].”

-Line 52: Change “…, site 1 affects antibody formation and host defense [15-17].” To “…, the antigenic site 1 induces a protective immunity [15-17].

-Line 200: Add “at”: “…the same survival rate at seven days post...”

-Line 221: Add “at”: “…was performed at 21 dpv.”

-Line 236: Remove “that”: “…or that TWN-R had a slightly higher relationship with…”

The English language is readable with a few grammatical errors and wrong word choices. However, the authors should explain more in detail and some topics should be re-organized chronologically.

Author Response

Thank you for reviewing the article. I tried to modified it based on your comments, so please check it out.

Reviewer 2 Report

Briefly, the research article submitted by Seong Yun Hwang and his colleagues aimed to test how well a recombinant epitope-based vaccination, made in mammalian monolayer cells, might provide protection against FMDV in mice and guinea pigs, pigs. Thus, Hwang et al. have attempted to evaluate the immunogenicity of the current synthesized recombinant ‘’novel’’ peptide candidate vaccine (TWN-JC strain) in comparison to ‘’previously studied’’ infectious clone (TWN-R) (as positive control) from unknow donor.  It is consensus of many scientists working on FMDV virus that development of broadly reactive vaccines which can confer immunity against multiple FMDV serotypes has yet been a difficult job. Similarly, issues related to DIVA vaccine, inefficient protection of the routine inactivated FMDV vaccines, thermal intolerability of vaccine components particularly at field level, and a short period of protection, all this are factors might be the reason why Hwang et al. focus to develop such vaccine designs. Recombinant epitope-based vaccine is very promising and certainly has tremendous safety, potency, and can assure a breadth of multiple serotypes or topotypes. To be honest, there is no question for the importance of such vaccine research design in the field of FMDV vaccinology.

General comments

This study has great potential for preventing and controlling FMDV outbreaks in endemic areas.  However, there are significant issues with the article as it is right now. As I've gone through the paper, from the abstract through the discussion section, the authors need to make significant changes. The abstract provided by the author is poorly written and does not effectively summarise the study as indicated by the title and detailed in the results. The purpose is not expressed clearly, which makes it less interesting to read. The experimental design (the number of experimental animals, age, sex, weight, breed etc)  is not properly explained, and the method as a whole is unclear. There have been several studies conducted on recombinant vaccination trials against FMDV serotype in Asia particularly in Korea, but none of these studies were described, addressed, or referred to in the introduction of the present study.

Line 17-21- To help the reader better understand your content, please reword the statement and reframe the idea. Line 20 of the sentence has a typo.

Line 57-62- at the first sentence of the paragraph, you say, "In a previous study," but it's not clear if that study was conducted by you or at your lab. Since the TWN-R FMDV strain serves as a baseline for your research. The point that the writer wishes to make in this paragraph is not clear. 

Line 65- 72- In (TWN-R), the location of the insertion of the chosen linear epitope (140-160) (construction of the recombinant) is ambiguous. It is unclear to me whether or not the primers have a restriction enzyme site at either end of the epitope coding sequence.

Line 71-73- Please explain in detail about the 3B region mutation? I am not sure if this variant is as part your previous studied strain.

Line 119-120- ‘’Mice were vaccinated via intramuscular injection, and mice in the 119 experimental group received a 1/10, 1/40, 1/160, or 1/640 dose of 15 μg/dose (pigs are vaccinated with 15 μg/dose’’ There is some fuzziness in this statement. Could you please elaborate?

Line 128-  ‘’APQA’’ This acronym was not defined in the preceding text, so its meaning is unclear.

Line 206- ‘’ However, mice inoculated with a 1/40 dose of TWN-R showed weight loss, whereas those inoculated with TWN-JC showed no weight loss’’ to understand their difference in the protective dose of the candidate ‘’novel’’ ,  TWN-R should be explained clearly in your study as this strain is used as a positive control.

Line 118- Have you considered how the (three adjuvants) in your prospective vaccine would affect its efficacy?

Line 285- Discussion and conclusion;

Although the authors conclude that serological conversion occurred in animals that were given a booster dosage, and that this is justified because the memory-related immunity was boosted by the addition of the JC epitope'', this immunological cascade mechanism is not clearly described in the discussion section. Do you think comparison of a single epitope insertion with multiple epitopes research article   

Author Response

(The authors gave the same response as above.)

Reviewer 3 Report

The manuscript entitled “Serological conversion through a second exposure to inactivated foot-and-mouth disease virus expressing a neutralizing epitope on the viral surface” by Hwang, et al., describes the generation of novel vaccine with objective to have wide range of coverage.

They have extended the previous work and developed novel TWN-JC strain by inserting the VP1 140–160 epitope of the O/SKR/Jin-58 cheon/2014 (JC) strain into the GH loop of VP1.

Authors have demonstrated robust immune response in guinea pig as well as pigs with additional epitopes against three topotypes of Serotype O.

Work is meticulously planned and executed. Results are appropriately presented and supported with statistical analysis and figures.

Authors have mentioned they performed 3D structural model analysis and results are presented as supplementary materials. However, readers may be interested about the results and discussion in main article, so authors may consider to include.

Author Response

(The authors gave the same response as above.)

Reviewer 4 Report

The manuscript described the immunogenicity and protection of an modified inactivated FMDV virus vaccine, in which a neutralizing epitope from the VP1 protein of a different topotype was inserted. Protection was evaluated in the mouse model after one immunization 7 days post vaccination. Guinea pig and  pigs were vaccinated twice 4 weeks apart and neutralizing antibodies were measured up to 4 weeks after the booster and compared against four different topotypes to evaluate homologous and heterologous antibody responses. Several major and minor comment below need to be addressed.

Major comments:

-The title doesn’t accurately reflect the findings of the study. “through a second exposure to ….” suggested that the hosts were primed by a “first exposure” and only the second exposure was to the antigen of interest. Also “expressing a neutralizing epitope on the viral surface” should specify the epitope is an additional neutralizing epitope, etc

-The author measured growth kinetics by measuring RNA copies, which doesn’t fully represent virus infectivity. Please provide growth kinetics by infectious titers (e.g TCID50)

-What is the rationale that in the mouse study, only one immunization was given and the challenge was performed only shortly (7 days) after the immunization? In contrast, a prime-boost regimen was used in Guinea pigs and pigs? The mouse study should be performed using similar schedule as the guinea pig and pig studies.

-Each result section was too short, the results were not sufficiently described

Minor comments:

-Line 20 correct “with0020”

-How much is the serological difference among SEA, ME-SA and cathay topotypes of type O? Why choosing JC VP1 epitope?

-Define SEA, ME-SA and Cathay

-Define BHK-T7-9, LFBK and ZZR-127 cell lines

-Define JC, VIT and BE

-How much of each adjuvant was used with the vaccine in each animal model?

-Provide vaccine dose as microgram instead of 1/10, 1/40,1/160 and 1/640. Update Table 1 accordingly

-What is the dose of the vaccine used in guinea pigs?

-Define SP-ELISA, it was not clearly described.

-Specify the strain of “homologous”, is it O1 manisa?

-The term of “serological relationship” is vague, please be more specific. Does it mean cross neutralizing?

Minor editing of english is needed to clearly describe the findings of this study.

Author Response

(The authors gave the same response as above.)

Reviewer 5 Report

Hwang et al, have looked to compare the serological responses between a parental vaccine strain, TWN-R, and a modified vaccine strain, TWN-JC, by replacing antigenic site 1 of TWN-R to match VP1 amino acids 140-160 of O/SKR/Jin- 58 cheon/2014 (JC) strain. The authors found that this novel vaccine candidate was able to effectively protect mice against three different topotypes of FMDV, similar to the parental strain, but with some increased efficacy. Additionally, the authors sought to investigate the serological response in both a guinea pig and pig model, seeing a trend towards increased antibody titres and increased viral neutralisation titres when using TWN-JC. Finally, the authors looked to determine the cause of this trend, and also saw a trend towards increased IFN-gamma and IL-12B. I believe this paper to be of interest to the community with some positive results, however, I feel that this paper has a number of further analyses required to determine the significance of the trends reported here before it can be accepted for publication.

Major Comments

Line 184-188 – I don’t believe the authors can use the term ‘confirmed’ for the change in predicted structure, especially as this is one predicted model. Furthermore, the authors do not add any further comment or analysis to this predicted structure and therefore does not currently add to the manuscript. If the authors wish to keep this figure, I recommend that they re-word their description and add further detail as to how the structure has changed and what this may mean, alternatively, I suggest that they could remove this figure.

Figure S2 – I believe that this adds something for the readership and should me moved into Figure 1 with slightly increased description. However, this assay appears to have a N=1, therefore I would like to see replicates of this data should it be included within the manuscript.

Figure 1 – The authors do not refer to Figure 1C or D within the text, they should either be referred to or removed from the manuscript.  Additionally, Figures 1A and B could be combined into 1A and C and D could become 1B.

Figure 2 – There are currently no statistical analyses here to prove the increased performance of TWN-JC. I suggest the authors add the analysis and comment on their significance prior to publication.

Line 206 – The authors refer to a 1/40 dose, however they do not describe what this 1/40 dose is in the text, instead only in the Figure legends does it become clear that they are describing the usual porcine vaccine dose. The authors should consider re-wording this part of the manuscript for increased clarity for the reader.

Figure 5 – Whilst I appreciate that the authors are trying to add another layer of analysis, I am unsure if this figure adds value to the manuscript. The authors haven’t provided clear rationale as to why Cathay VN titers were used to normalise the VN data to look at the ability to produce neutralising antibodies against each of the Topotypes. As I believe the best readout would be the VN titre for each virus? Additionally, when looking at the ability to boost, I think looking at the fold change between the prime and boost titres would make for more compelling data than normalising to a specific topotype which would inherently skew the data. Ideally, the authors would provide clear rationale for this analysis in the manuscript or consider another analysis to compare the serological responses against each topotype.  

Figure 6 – There is no statistical analysis of this data, the authors should conduct the appropriate statistical analysis to back their claim.

Figure 3, 4, 5 & 6 – Could the authors comment on how many animals’ sera they have used in each experiment within the legend. Additionally, showing the individual data points within a clear bar would allow for greater understanding of the data for the readership.

Minor Comments

Line 20 – 0020 to be removed from the middle of the line

Line 23-25 – The conclusion reads okay until the final part of the sentence, where the authors add an extra bit of information regarding the induction of IFN gamma and IL-12B. To improve the impact and the take home message, I think referring to the cytokine induction earlier in the abstract and ending on the clear differences in serological responses to the two different vaccines would highlight the conclusions of this work to the readership clearly.

Figure 2 – The figure legend doesn’t include 2C -  The authors should edit the figure legend to reflect the contents of A, B & C.

Discussion – I think some of the conclusions of the discussion will need to readdressed following the analysis and suggested changes suggested in the major points.

Author Response

(The authors gave the same response as above.)

Round 2

Reviewer 1 Report

Upon evaluation of the revised manuscript and Response to reviewer, there are important points that need attention.

1.      Mouse models for FMDV studies were previously reviewed (Habiela et al., J Gen Virol. 2014 Nov; 95(Pt 11): 2329–2345. doi: 10.1099/vir.0.068270-02014) and reported by you and your team (Lee et al., Vaccine. Volume 34, Issue 33, 19 July 2016, Pages 3731-3737); however, FMD vaccine efficacy test using a mouse model has not been well established. Although this study demonstrated that TWN-JC vaccine could induce robust immune responses against different FMDV topotypes in the prime-boost regimen, number of mice in each experimental group were not equal and too small to establish the standard mouse protective value 50 (mPD50). Therefore, the mouse protective value 50 or mPD50 should be replaced by survival rate.

2.      In the 3D structure (Figure S1), the insertion of FMDV JC strain (O/SEA/Mya98) VP1 B cell epitope in the G-H loop of TWN-R (Cathay topotype) resulted in an unusual flexible G-H loop, which might affect the virus assembly and the binding between the virus and receptor. The authors stated that this issue was proved by the ability of the recombinant TWN-JC to replicate in ZZ-R 127, LFBK and BHK21 cells and form particles resemble TWN-R as examined by EM. However, during the cloning process, it was possible that the clone with and without this B-cell epitope inserted sequence were mixed populations. Therefore, the authors must provide the following evidence and explanation.

2.1   DNA sequencing of the VP1 region of the TWN-JC stock used for the immunization experiments to show that it contains the JC VP1 B cell epitope.

2.2   In the discussion, the authors should discuss how TWN-JC enter the cells. One explanation is that it used alternative receptor(s) such as heparan sulfate or else.

3.      The authors should discuss why both TWN-R and TWN-JC induced better immune responses against ME-SA (O/VIT/2013) than the homologous viruses in the mouse experiment.

4.      Correction

-Line 257, topic of Table 1: change “…3 topotype challenge viruses.” to “   3 topotypes of challenge viruses.”

-Line 348, Results of Figure 5: “…and TWN-JC had a similar…; however,” to “…and TWN-JC had a similar SNT level.”

-Line 362, the last sentence of Figure 5 caption: “…fold change was normalized boost VN titer to prime VN titer.” to “…fold change of the boost VN titer was normalized to prime VN titer.”

-Line 367, under topic 3.5: “Compared to Normalized with pre-vaccination serum” to By normalized with pre-vaccination serum”

English writing is fine. Carefully editing is required.

Author Response

Thank you for reviewing this article, your comments made the article much better. We have tried to revise this article as much as possible according to your opinions. We hope for a positive and quick reply.

Thank you for reviewing this article, your comments made the article much better.

reviewer A.

  1. Mouse models for FMDV studies were previously reviewed (Habiela et al., J Gen Virol. 2014 Nov; 95(Pt 11): 2329?2345. doi: 10.1099/vir.0.068270-02014) and reported by you and your team (Lee et al., Vaccine. Volume 34, Issue 33, 19 July 2016, Pages 3731-3737); however, FMD vaccine efficacy test using a mouse model has not been well established. Although this study demonstrated that TWN-JC vaccine could induce robust immune responses against different FMDV topotypes in the prime-boost regimen, number of mice in each experimental group were not equal and too small to establish the standard mouse protective value 50 (mPD50). Therefore, the mouse protective value 50 or mPD50 should be replaced by survival rate.

A : In response to your opinions, we have decided to remove table 1 and replace it with the survival rate in Figure 2B.

  1. In the 3D structure (Figure S1), the insertion of FMDV JC strain (O/SEA/Mya98) VP1 B cell epitope in the G-H loop of TWN-R (Cathay topotype) resulted in an unusual flexible G-H loop, which might affect the virus assembly and the binding between the virus and receptor. The authors stated that this issue was proved by the ability of the recombinant TWN-JC to replicate in ZZ-R 127, LFBK and BHK21 cells and form particles resemble TWN-R as examined by EM. However, during the cloning process, it was possible that the clone with and without this B-cell epitope inserted sequence were mixed populations. Therefore, the authors must provide the following evidence and explanation.

2.1   DNA sequencing of the VP1 region of the TWN-JC stock used for the immunization experiments to show that it contains the JC VP1 B cell epitope.

A : we modified Figure S1. Following your advice, VP1 sequencing data of the viruses used in the antigen purification process has been added.

2.2   In the discussion, the authors should discuss how TWN-JC enter the cells. One explanation is that it used alternative receptor(s) such as heparan sulfate or else.

A : Thank you for your good comments, there are two RGD motif in TWN-JC by inserting JC epitope, and it is known that the RGD motif interacts with the Integrin receptor. We think the difference in mechanism (virus entry) due to the presence of two RGD motifs can be used as a good idea for next study.

  1. The authors should discuss why both TWN-R and TWN-JC induced better immune responses against ME-SA (O/VIT/2013) than the homologous viruses in the mouse experiment.
    A : we have taken your advice and added it to the manuscript discussion.

As a result of mice challenge test, the reason why the mouse challenge inoculation results showed a higher survival rate in the group challenged with VIT than in the group TWN or JC in both TWN-R and TWN-JC is that the Pan-Asia lineage among ME-SA is effective against most type O viruses. It is thought to show a tendency to be highly reactive. (reference 37)

  1. Correction

-Line 257, topic of Table 1: change “…3 topotype challenge viruses.” to “   3 topotypes of challenge viruses.”

A : In response to your opinions, we have decided to remove table 1 and replace it with the survival rate in Figure 2B.

-Line 348, Results of Figure 5: “…and TWN-JC had a similar…; however,” to “…and TWN-JC had a similar SNT level.”

A : we rewrote Results of Figure 5: “…and TWN-JC had a similar…

-Line 362, the last sentence of Figure 5 caption: “…fold change was normalized boost VN titer to prime VN titer.” to “…fold change of the boost VN titer was normalized to prime VN titer.”

A : A : we changed change “…fold change was normalized boost VN titer to prime VN titer.” to “…fold change of the boost VN titer was normalized to prime VN titer.”

-Line 367, under topic 3.5: “Compared to Normalized with pre-vaccination serum” to By normalized with pre-vaccination serum”

A : we rewrote “Compared to Normalized with pre-vaccination serum” to By normalized with pre-vaccination serum”

Reviewer 2 Report

I have gone through the revised paper ’Serological conversion through a second exposure to inactivated foot-and-mouth disease virus expressing JC epitope on the viral surface‘after the authors made a major revision. Hence, I can confirm that the authors made considerable changes in the revised manuscript in accordance to my comments and suggestions. I believed that all my comments are addressed in the current version of the article.

Author Response

Thank you for reviewing this article, your comments made the article much better. We have tried to revise this article as much as possible according to your opinions. We hope for a positive and quick reply.

Thank you for reviewing this article, your comments made the article much better.

Review B.

I have gone through the revised paper ’Serological conversion through a second exposure to inactivated foot-and-mouth disease virus expressing JC epitope on the viral surface‘after the authors made a major revision. Hence, I can confirm that the authors made considerable changes in the revised manuscript in accordance to my comments and suggestions. I believed that all my comments are addressed in the current version of the article.

A : Thank you for your good advice, which has resulted in a much better article. We hope things will work out all right.

Reviewer 5 Report

I would like to thank the authors for taking the time to consider the review points and answer them individually. I have only one suggestion which I believe will improve the paper listed below: 

Figure S2 – As this is a key result which is described in the text alongside the TEM images, I believe this figure would be of interest to the readership as part of the main text. Therefore, if possible the authors should move this figure to Figure 1. Additionally, I appreciate the authors adding error bars to the figure, if they could also add the number of replicates to the figure legend, that would be helpful to the readership.

Otherwise, I would like to congratulate the authors on their manuscript.

Author Response

Thank you for reviewing this article, your comments made the article much better. We have tried to revise this article as much as possible according to your opinions. We hope for a positive and quick reply.
